# Mutations in the HBV PreS/S gene related to hepatocellular carcinoma in Vietnamese chronic HBV-infected patients

Nguyen Thi Cam Huong [1,2]☯*, Nguyen Quang Trung[1,2]☯, Bac An Luong[3], Duong Bich Tram[3], Hoang Anh Vu[3], Hoang Huu Bui[2], Hoa Pham Thi Le[1,2]

**1** Department of Infectious Diseases, University of Medicine and Pharmacy at Ho Chi Minh City, Ho Chi Minh City, Vietnam, **2** Department of Gastroenterology, University Medical Center, Ho Chi Minh City, Vietnam, **3** Center for Molecular Biomedicine, University of Medicine and Pharmacy at Ho Chi Minh City, Ho Chi Minh City, Vietnam

☯ These authors contributed equally to this work.
* dr_camhuong@ump.edu.vn

**Data Availability Statement:** All relevant data are within the paper and its Supporting Information files.

## Abstract

### Background

Chronic hepatitis B virus (CHB) infection is a major health problem and leading cause of hepatocellular carcinoma (HCC) worldwide. Several point and deletion mutations on the *PreS/S* gene have been intensively considered associated with HCC. This study aimed to describe the characteristics of HBV *PreS/S* mutations in Vietnamese CHB-infected patients and their association with HCC.

### Methods

This cross-sectional study was conducted from 02/2020 to 03/2021, recruited Vietnamese CHB-infected patients with HBV-DNA >3 $\log_{10}$-copies/mL and successful *PreS/S* gene sequencing. Mutations were detected by direct Sanger sequencing.

### Results

247 CHB-infected patients were recruited, characterized by 68.8% males, 54.7% HBV genotype B, 57.5% HBeAg positive, 23.1% fibrosis score ≥F3 and 19.8% HCC. 61.8% amino acid replacements were detected throughout the *PreS1/PreS2/S* genes. The most common point-mutations included N/H51Y/T/S/Q/P (30.4%), V68T/S/I (44.9%), T/N87S/T/P (46.2%) on *PreS1* gene; T125S/N/P (30.8%), I150T (42.5%) on *PreS2* gene; S53L (37.7%), A184V/G (39.3%), S210K/N/R/S (39.3%) on *S* gene. The rates of case(s) with any point-mutation on the Major Hydrophylic Region (MHR) and the "a" determinant region were 63.6% and 39.7%, respectively. Most of *S* point-mutations were presented with low rates such as T47A/E/V/K (9.3%), P120S/T (8.5%), G145R (2%). On multivariable analysis, males (OR = 4.51, 95%CI 1.78–11.4, p = 0.001), age≥40 (OR = 5.5, 95%CI 2.06–14.68, p = 0.001), W4P/R/Y on PreS1 (OR = 11.56, 95%CI 1.99–67.05, p = 0.006) and 4 S point-mutations as: T47A/E/V/K (OR = 3.67, 95%CI 1.19–11.29, p = 0.023), P120S/T (OR = 3.38, 95%

**Funding:** This study was funded for the PreS/S gene Sequencing by the grant of research from the University of Medicine and Pharmacy at Ho Chi Minh city. The funders had no role in study design, data collection and analysis, decision to publish, or preparation of the manuscript.

**Competing interests:** The authors have declared that no competing interests exist.

CI 1.09–10.49, p = 0.035), S174N (OR = 29.73, 95%CI 2.12–417.07, p = 0.012), P203R (OR = 8.45, 95%CI 1.43–50.06, p = 0.019) were associated with HCC.

## Conclusions

We detected 61% amino acid changes on *PreS/S* regions in Vietnamese CHB patients. One point-mutation at amino acid 4 on *PreS1* gene and 4 point-mutations at amino acids 47, 120, 174, and 203 on *S* gene were associated with HCC. Further investigations are recommended to further clarify the relationship and interaction between mutations in HBV genome and HCC progression.

## Introduction

Chronic HBV (CHB) infection affects 296 million people worldwide in 2019 [1], and has been considered as a major global health problem. CHB infection is the leading cause of liver cirrhosis and contributes over 50% of hepatocellular carcinoma (HCC) [2]. Vietnam locates in the HBV-high-prevalence area in Asia [3] which has the high incidence of HBV related-HCC [2]. It was reported that 62.3% of HCC cases and 81.3% of advanced HCC cases in Vietnam were HBV infected [4, 5].

HBV belongs to the Hepadnaviridae family with an incomplete double strand DNA genome, which carries 4 overlapped open reading frames, coding for 4 proteins PreS/S, pre-Core/Core, Pol, and X. HBsAg proteins are trancsripted from *PreS/S* open reading frame that consists of 3 surface proteins: Small (S), Medium (M) and Large (L). The S protein that drives the releasing of viral particles consists of 227 amino acids (aa). The M protein that enriches the secretion of virion contains an extra N-terminal extension of 55 aa. The L protein that is involved in the interaction with core particles in the packaging of virion at the endoplasmic reticulum (ER) has a further N-terminal extension of 108 or 119 aa–depending on genotypes [6]. In the absence of L protein, S protein is secreted alone as noninfectious subviral particles. L protein can suppress the subviral particle secretion depending on the L/S protein ratio. During natural HBV infection subviral particles outnumber virions by a factor of 1000:1. The principal epitopes of HBsAg mainly locate in the "a" determinant (aa 124–147) in the major hydrophilic region (MHR) induces the neutralized B cell responses.

Mutations in *PreS/S* gene result in deletion of surface proteins or synthesis of varieties of truncated proteins. Especially, mutations on the C-terminal region (aa 179–226) of *S* gene contribute in retention of HBsAg within the hepatocytes ER [7, 8], activate multiple oncogenic signal pathways, promote the growth of hepatocytes and eventually lead to HCC development. Multiple scientific evidences had proved *PreS* mutations as prediction markers for HCC development and recurrence of HBV-related HCC [9–12]. The relationship between *PreS/S* gene mutations and HCC were studied distinctly on the *PreS* region [13]. Mutations at T53C, *PreS* deletions, *PreS2* start codon, C7A, A2962G, C2964A and C3116T in the *PreS* region have been proved that significantly increase risk of HCC [14, 15]. Wang *et al.* (2006) had concluded that the retention of L antigens from the *PreS* mutants caused ER stress, induced oxidative DNA damage, and resulted in genomic instability. The L antigens from the *PreS* mutants are attributed to the upregulation cyclooxygenase-2 and cyclin A, and promotion of cell cycle and hepatocytes proliferation [16].

Mutations on the *S* gene in Vietnamese CHB patients had been described in the earliest study since 2012. Dunford *et al.* (n = 187) had reported a rate of 31% cases with point-

mutation in the immunodominant 'a' region, especially two major vaccine escape mutations with minor rates as G145A/R (2.2%) and P120L/Q/S/T (5.3%) had been detected [17]. The mutation of *PreS* deletion with a rate of 20% was reported from Matsuo *et al.* (2017) in Vietnamese CHB-infected patients [18]. Bui TTT *et al.* (2017) described more concretely about point-mutations (N38E 71.9%, N38K 71.1%, A60V/E 100% on the *PreS1* region, L126T/S 77% on the *PreS2* and N3S 27.4% on the *S* region) [19]. In a multicenter study on 660 patients from China, Korea and Vietnam, Kim *et al.* (2018) [20] had reported 237 amino acid mutations in the MHR on the *S* region. There was not any report on mutations and their association with HCC on the *PreS/S* gene in Vietnamese patients.

In this study, we described the characteristics of HBV *PreS/S* gene mutations in Vietnamese patient with CHB and analyzed the relations of these mutations with HCC.

## Materials and methods

### Study design and population

The cross-sectional study had been conducted at the Hepatology Clinic of University Medical Center (UMC), Ho Chi Minh city from February 2020 to March 2021. There were 293 male and female CHB patients paticipated in this study, who met the inclusion criteria of being older than 18 years old, had HBsAg positive more than 6 months, no previous nucleos(t)ide analogues treatment (NAs) and HBV DNA >3 $\log_{10}$ copies/mL. Their serum samples were extracted from 4 mL blood, stored in -80 celcius degree for *PreS/S* gene sequencing. Serum samples of 247 patients that had been successfuly sequenced the HBV *PreS/S* gene were analyzed for the final results. Among them, there were 212 CHB patients whose serum samples had been stored during 2014–2016 and 35 CHB patients were recruited in 2020–2021.

### Variables and measurements

Personal characteristics, times from diagnosis of CHB infection and HBV markers were collected from the hospital electronic database. HCC was defined for cases with tumor detected on abdominal ultrasound, serum alpha-fetoprotein (AFP) >20 ng/mL and was confirmed on abdominal CT scan with focal lesions with early arterial phase enhancement and rapid "washout" in venous phase [21]. Cirrhosis was defined as having signs of portal hypertension (splenomegaly, ascites, vascular collaterals on abdominal wall, esophageal varices or portal hypertensive gastropathy on gastroscopy) and signs of insufficiency of liver function (palmar erythema, vascular spiders, low concentration of albumin (<35 g/dL), elevation of the international normalized ratio (INR >1.1), thrombocytopenia (<160,000/mm$^3$)) as well as irregularity of hepatic surface on ultrasonography or F3 on Metavir score using Acoustic radiation force impulse (ARFI) measurement [22].

HBsAg quantification (Elecsys HBsAgII Quant-Roche kit), HBeAg (Cobas-Roche kit) and HBV DNA quantification (using the AccuPid HBV Quantification kit (KT-Biotech)) with limits of detection ≥300 copies/mL, linear range: 300 to $10^8$ copies/mL) were performed at the University Medical Center laboratory. HBV genotype (B or C) was determined based on the sequence of *S* gene.

### *PreS/S* mutation analyzed by sanger sequencing

*PreS/S* mutations were analyzed at Center for Molecular Biomedicine of University of Medicine and Pharmacy (UMP) at Ho Chi Minh City. HBV DNA was extracted from serum using the GeneJet$^{TM}$ Viral DNA and RNA Purification kit (Thermo Fisher Scientific, Waltham, MA, USA) according to the manufacturer's protocol. Two sets of overlapping primers were then

used to amplify the whole *PreS/S* region of HBV with TaKaRa Taq^TM HotStart Polymerase (Takara Bio, Shiga, Japan). Primers for the *PreS1/PreS2* region were: FA2-L (5'- TTGAGA-GAAGTCCACCACGAG-3') and FA2-R (5'-GCGTCGCAGAAGATCTCAAT-3'); S region were FA3-L (5'-CTGCTGGTGGCTCCAGTT-3'); FA3-R (5'-GCCTTGTAAGTTGGCGA-GAA-3'). PCR involved initial denaturation at 98˚C for 3 min followed by 45 cycles of 98˚C for 10 sec, 58˚C for 30 sec, and 72˚C for 60 sec, with a final elongation of 72˚C for 2 min. PCR products were checked for size and purity using 1.5% agarose gel electrophoresis. PCR product was purified enzymatically using the ExoSAP-IT^TM PCR Product Cleanup Reagent (Thermo Scientific, Waltham, MA) to remove excess primers and dNTPs before Sanger sequencing using the BigDye Terminator v3.1 Kit and the ABI 3500 Genetic Analyzer (Applied Biosystems, Foster City, CA). PCR fragment was sequenced and analyzed in both directions. The sequences were compared to the reference sequence of genotype B (GenBank_AB073846) and genotype C (GenBank_X04615) using the CLC Main Workbench software (Qiagen, Germany).

## Statistics

The SPSS 25.0 software was used to analyze the data. Percentage was used to present the rates of point-mutations, the rates of possessing at least one mutation and the rates of genes deletion on each region. The Chi-square test (or Fisher exact test) was used to compare the distributions of mutations among groups with or without HCC. Multivariable analysis with logistic regression was used to find out factors related to HCC. Two-side p value of <0.05 was considered statistically significant.

## Ethics considerations

The study was done based on the Declaration of Helsinki. Stored serum samples and all variables included in the study was approved by the Ethics Committee of the University of Medicine and Pharmacy at Ho Chi Minh City (Reference number: 119/HDDD). Informed consents were obtained from all participants prior to the study.

## Results

### Characteristics of the study population

The study included 247 CHB patients, 68.8% were males, 57.9% were older than 40 years of age and older. 57.5% were positive with HBeAg marker, 83% had HBV DNA $\geq 5 \log_{10}$ copies/mL, 54.7% genotype B. 23.1% were with liver fibrosis and 19.8% were with HCC (*Table 1*).

**Table 1. Characteristics of the study population (n = 247).**

| Characteristics | | n (%) |
|---|---|---|
| Sex (male) | | 170 (68.8) |
| Age group $\geq$40 | | 143 (57.9) |
| HBeAg positive | | 142 (57.5) |
| HBV DNA $\geq$5 ($\log_{10}$ copies/mL) | | 205 (83) |
| Genotype B (n = 245) | | 135 (54.7) |
| Fibrosis $\geq$F3 | | 57 (23.1) |
| HCC | | 49 (19.8) |

## Characteristics of the detected mutations on the *PreS1*, *PreS2* and *S* genes in overall population and the HCC subgroups

There were 248/401 (61.8%) amino acid replacements that were detected throughout the *PreS1/PreS2/S* genes on the overall populations. In the *PreS1* region, 57.1% replacements (68/119) were found. The mutations with a rate over 30% were N51Y/T/S/Q/P 30.4%, V68T/S/I 44.9%, and T/N87S/T/P 46.2%; from 10 to <30% were Q10K/H/R (16.2%), H48Y/N (13.8%), E/D54A/N (25.1%), I/N56H/W (25.9%), K57Q/K (25.1%), A60V (25.5%), D/A62S/T (25.1%), G73S/N (24.3%), and V95A (24.3%); and from 5 to <10% were G35R/K (8.5%), and I84V/M/L (7.7%). These above mutations were not differently distributed among non-HCC and HCC group. Interestingly, most of amino acid changes (54/68) in the preS1 were presented with the rates <5%. Among them, 4 point-mutations W4P/R/Y, S5L/T, A90T/S/G, and L108V/I were observed with the significantly higher rates in the HCC group compared to non-HCC group: W4P/R/Y (12.2% vs 2%, p = 0.005), S5L/T (6.1% vs 1%, p = 0.055), A90T/S/G (6.1% vs 0.5%, p = 0.026) and L108V/I (6.1% vs 0.5%, p = 0.026) (*Table 2*).

In the *PreS2* region, 41/55 (74.5%) amino acid changes were detected. The point-mutations with the rates of over 30% were T125S/N/P (30.8%), I150T (42.5%), and V158A (36%); from 10 to <30% of population were M120V/I (11.3%) and F141V/L/I (11.3%); and from 5 to <10% of population were Q121R/K (5.3%), T164I/D/S (6.1%), and F165S/L (5.7%). As same as the *PreS1* region, most of amino acid changes on the *PreS2* region (34/41 sites) were presented at a rate <5% (Table 3). Only F141V/L/I was found with a significantly higher rate in the HCC group (18.4% vs 9.6%, p = 0.08).

The *PreS1* deletion was detected in 27.5% (68 patients), and the *PreS2* deletion was observed in 16.2% (40 patients) (*Tables 2 and 3*).

In the *S* region, 61.2% amino acid changes were detected (139/227). The rate of cases with at least one point-mutation detected on the "a" determinant region (aa 124–148) was 39.7% and on the MHR region (aa 100–160) was 63.6%. The HCC group had significant higher rate of possessing ≥1 point-mutation on the MHR region (79.6% vs 59.6%, p = 0.009).

The point-mutations on the *S* gene that owned the rates of >30% of the population were: S53L (37.7%), A184V/G (39.3%), and S210K/N/R/S (39.3%); from 15 to <30% were L21S (29.1%), G44E/V (18.6%), I126T/N/S (21.1%), and M198I/M (18.2%); and from 5 to <15% were V14A/G/Q (10.1%), N40S/K (6.9%), T47A/E/V/K (9.3%), P/L49R/H (5.7%), P62Q/L (9.7%), C76Y/T/W (10.5%), Y100C/F (5.3%), P120S/T (8.5%), R122K (8.9%), M133T/S/L/I (7.7%), Y161F/S (10.1%), T189I (5.3%), S204R/N (10.1%), I208T/S (5.7%), L213I/M (7.3%), and V224A (11.7%). Half of the amino acid changes (116/227) on the *S* region had the detection rates <5% and most of them were not related to HCC. Exceptionally, 13 *S* mutations presented the higher distributions (p<0.1) in the HCC group: F20S (8.2% vs 1%, p = 0.015), D33G (4.1% vs 0%, p = 0.039), (T47A/E/V/K (18.4% vs 7.1%, p = 0.025), R79H (6.1% vs 0%, p = 0.007), L88P (4.1% vs 0%, p = 0.039), P120S/T (on the MHR region, 16.3% vs 6.6%, p = 0.042), G145R (6.1% vs 1%, p = 0.055), S174N (6.1% vs 0.5%, p = 0.026), V190A (6.1% vs 0.5%, p = 0.026), P203R (8.2% vs 2%, p = 0.052), Y206H/F/C (6.1% vs 1.5%, p = 0.094), L209V/S/G (6.1% vs 1%, p = 0.055) and F212Y/L/C (8.2% vs 0.5%, p = 0.006), (*Table 4*).

## The point-mutations on the *Pres1/Pres2/S* genes related to HCC–multivariable analysis

Nineteen point-mutations that distributed differently (p<0.1) among the HCC and non-HCC groups were checked to remove their interactions using multivariable analysis (Tables 2–4). Six point-mutations remained related to HCC. They were one mutation on the *PreS1* region: W4P/R/Y (OR = 5.48, 95%CI 1.32–22.83) and 5 mutations on the *S* region: F20S (OR = 9.72,

**Table 2. Distribution of point and deletion mutations on the *PreS1* gene (n = 247).**

| PreS1 (aa 1–119) | Overall population n (%) | HCC n (%) | | p<sup>a</sup> |
|---|---|---|---|---|
| | | Yes (n = 49) | No (n = 198) | |
| **Point-mutations** | | | | |
| G2R/G | 1 (0.4) | 0 | 1 (0.5) | 1 |
| W4P/R/Y | 10 (4) | **6 (12.2)** | **4 (2.0)** | **0.005<sup>b</sup>** |
| S5L/T | 5 (2) | **3 (6.1)** | **2 (1.0)** | **0.055<sup>b</sup>** |
| S6A | 3 (1.2) | 1 (2.0) | 2 (1.0) | 0.49 |
| K7N | 1 (0.4) | 0 | 1 (0.5) | 1 |
| P8T | 1 (0.4) | 1 (2.0) | 0 | 0.2 |
| Q10K/H/R | 40 (16.2) | 8 (16.3) | 32 (16.2) | 0.98 |
| T14I/T | 1 (0.4) | 0 | 1 (0.5) | 1 |
| S17F/A | 4 (1.6) | 1 (2.0) | 3 (1.5) | 1 |
| P19S | 1 (0.4) | 0 | 1 (0.5) | 1 |
| F25L | 1 (0.4) | 1 (2.0) | 0 | 0.2 |
| D27G/S | 33 (13.4) | 6 (12.2) | 27 (13.6) | 0.8 |
| I31T | 1 (0.4) | 1 (2.0) | 0 | 0.2 |
| P32L | 3 (1.2) | 0 | 3 (1.5) | 1 |
| A33F/L | 2 (0.8) | 1 (2.0) | 1 (0.5) | 0.36 |
| F34Y | 1 (0.4) | 0 | 1 (0.5) | 1 |
| G35R/K | 21 (8.5) | 3 (6.1) | 18 (9.1) | 0.78 |
| S38T | 1 (0.4) | 0 | 1 (0.5) | 1 |
| N/E39K/G/D/A | 11 (4.5) | 2 (4.1) | 9 (4.5) | 1 |
| N40Y/T | 2 (0.8) | 0 | 2 (1.0) | 1 |
| D42N | 1 (0.4) | 0 | 1 (0.5) | 1 |
| D44H/N | 3 (1.2) | 0 | 3 (1.5) | 1 |
| L45R/F | 10 (4.0) | 3 (6.1) | 7 (3.5) | 0.42 |
| H48Y/N | 34 (13.8) | 3 (6.1) | 31 (15.7) | 0.08 |
| D50N/E | 2 (0.8) | 1 (2.0) | 1 (0.5) | 0.36 |
| N/H51Y/T/S/Q/P | 75 (30.4) | 15 (30.6) | 60 (30.3) | 0.97 |
| E/D54A/N | 62 (25.1) | 9 (18.4) | 53 (26.8) | 0.23 |
| A55S | 1 (0.4) | 0 | 1 (0.5) | 1 |
| I/N56H/W | 64 (25.9) | 14 (18.6) | 50 (25.3) | 0.64 |
| K57Q/K | 62 (25.1) | 13 (26.5) | 49 (24.7) | 0.80 |
| G59A | 1 (0.4) | 1 (2.0) | 0 | 0.2 |
| A60V | 63 (25.5) | 14 (28.6) | 49 (24.7) | 0.58 |
| D/A62S/T | 62 (25.1) | 15 (30.6) | 47 (23.7) | 0.32 |
| P65T | 1 (0.4) | 0 | 1 (0.5) | 1 |
| F67V/L | 2 (0.8) | 0 | 2 (1.0) | 1 |
| V68T/S/I | 111 (44.9) | 16 (32.7) | 95 (48) | **0.053<sup>a</sup>** |
| P70S | 1 (0.4) | 0 | 1 (0.5) | 1 |
| G73S/N | 60 (24.3) | 11 (22.4) | 49 (24.7) | 0.74 |
| L74M/V | 6 (2.4) | 1 (2.0) | 5 (2.5) | 1 |
| L75V/M | 5 (2.0) | 0 | 5 (2.5) | 0.59 |
| W77R/G | 5 (2.0) | 1 (2.0) | 4 (2.0) | 1 |
| S78N | 3 (1.2) | 1 (2.0) | 2 (1.0) | 0.49 |
| Q80L/H | 3 (1.2) | 1 (2.0) | 2 (1.0) | 0.49 |
| A81T | 3 (1.2) | 1 (2.0) | 2 (1.0) | 0.49 |
| Q82L | 1 (0.4) | 0 | 1 (0.5) | 1 |

*(Continued)*

**Table 2.** (Continued)

| PreS1 (aa 1–119) | Overall population n (%) | HCC n (%) | | p[a] |
|---|---|---|---|---|
| | | Yes (n = 49) | No (n = 198) | |
| I84V/M/L | 19 (7.7) | 3 (6.1) | 16 (8.1) | 0.77 |
| L85I/H/F | 3 (1.2) | 1 (2.0) | 2 (1.0) | 0.49 |
| T86A/S | 6 (2.4) | 2 (4.1) | 4 (2.0) | 0.34 |
| T/N87S/T/P | 114 (46.2) | 18 (36.7) | 96 (48.5) | 0.14 |
| V88I/M/L | 5 (2) | 2 (4.1) | 3 (1.5) | 0.26 |
| P89R | 1 (0.4) | 0 | 1 (0.5) | 1 |
| A90T/S/G | 4 (1.6) | **3 (6.1)** | **1 (0.5)** | **0.026[b]** |
| A91T/P | 8 (3.2) | 1 (2.0) | 7 (3.5) | 1 |
| P92L | 1 (0.4) | 0 | 1 (0.5) | 1 |
| P94T/S | 5 (2.0) | 2 (4.1) | 3 (1.5) | 0.26 |
| V95A | 60 (24.3) | 12 (24.5) | 48 (24.2) | 0.97 |
| T97I/T/A | 4 (1.6) | 0 | 4 (2.0) | 0.59 |
| N98T/K/I | 3 (1.2) | 0 | 3 (1.5) | 1 |
| S101T/L | 2 (0.8) | 0 | 2 (1.0) | 1 |
| G102R/K | 2 (0.8) | 0 | 2 (1.0) | 1 |
| R103T | 1 (0.4) | 0 | 1 (0.5) | 1 |
| Q104R/K | 4 (1.6) | 2 (4.1) | 2 (1.0) | 0.18 |
| L108V/I | 4 (1.6) | **3 (6.1)** | **1 (0.5)** | **0.026[b]** |
| S109T | 11 (4.5) | 2 (4.1) | 9 (4.5) | 1 |
| R113T | 1 (0.4) | 0 | 1 (0.5) | 1 |
| D114E | 1 (0.4) | 0 | 1 (0.5) | 1 |
| T115S/C | 6 (2.4) | 0 | 6 (3.0) | 0.6 |
| Q118L | 1 (0.4) | 0 | 1 (0.5) | 1 |
| A119V | 1 (0/4) | 0 | 1 (0.5) | 1 |
| PreS1 deletion | 68 (27.5%) | 15 (30.6) | 53 (26.8) | 0.59 |

percentage by column, [a] Chi-square test, [b] Fisher Exact test.

95%CI 1.55–61.06), T47A/E/V/K (OR = 2.91, 95%CI 1.04–8.13), P120S/T (OR = 4.26, 95%CI 1.58–11.52), S174N (OR = 18.21, 95%CI 1.77–187.65), and P203R (OR = 9.72, 95%CI 1.55–61.06) (Table 5).

Personal and viral characteristics were also evaluated for their confounding effects on the correlation between mutations and HCC. Five basal characters that had different distributions in HCC and non-HCC groups were male (83.7% vs 65.2%, p = 0.012), age group >40 (85.7% vs 51%, p<0.001), HBeAg negative (61.2% vs 37.9%, p = 0.003), HBV DNA <5 $\log_{10}$-copies/mL (46.9% vs 9.6%, p<0.001), and liver fibrosis of >F3 (38.8% vs 19.2%, p = 0.004) (S2 Table).

The final multivariable analysis for related factors to HCC had included gender, age group, HBeAg marker and 6 mutations (PreS1 W4P/R/Y and five S point-mutations as F20S, T47A/E/V/K, P120S/T, S174N, P203R) on Table 5. The result found eight variables that composed of males (OR = 4.51, 95%CI 1.78–11.4, p = 0.001), age ≥40 (OR = 5.5, 95%CI 2.06–14.68, p = 0.001), HBeAg negative (OR = 2.46, 95%CI 1.1–5.53, p = 0.029) and 5 point-mutations such as W4P/R/Y (OR = 11.56, 95%CI 1.99–67.05, p = 0.006), T47A/E/V/K (OR = 3.67, 95%CI 1.19–11.29, p = 0.023), P120S/T (OR = 3.38, 95%CI 1.09–10.49, p = 0.035), S174N (OR = 29.73, 95%CI 2.12–417.07, p = 0.012) and P203S (OR = 8.45, 95%CI 1.43–50.06, p = 0.019) had significant relations with HCC (Table 6). S174N (S) had the highest OR (29.73) with positive predictive value (PPV), negative predictive value (NPV), sentitivity (SEN),

**Table 3. Distribution of point and deletion mutations on the *PreS2* gene (n = 247).**

| *PreS2 (aa 120–174)* | Overall population n (%) | HCC n (%) | | p [a] |
|---|---|---|---|---|
| | | Yes (n = 49) | No (n = 198) | |
| **Point-mutations** | | | | |
| M120V/I | 28 (11.3) | 7 (14.3) | 21 (10.6) | 0.47 |
| Q121R/K | 13 (5.3) | 1 (2.0) | 12 (6.1) | 0.47 |
| W122R | 3 (1.2) | 1 (2.0) | 2 (1.0) | 0.49 |
| N123T/K | 3 (1.2) | 1 (2.0) | 2 (1.0) | 0.49 |
| S124T | 8 (3.2) | 1 (2.0) | 7 (3.5) | 0.29 |
| T125S/N/P | 76 (30.8) | 12 (24.5) | 64 (32.3) | 0.29 |
| T126N/I/A | 3 (1.2) | 1 (2.0) | 2 (1.0) | 0.49 |
| H128L | 4 (1.6) | 1 (2.0) | 3 (1.5) | 1 |
| Q129K | 1 (0.4) | 1 (2.0) | 0 | 0.2 |
| A130T/N | 12 (4.9) | 3 (6.1) | 9 (4.5) | 0.71 |
| L/Q132I/H | 5 (2.0) | 1 (2) | 4 (2) | 1 |
| D133N | 1 (0.4) | 0 | 1 (0.5) | 1 |
| P134H/T | 2 (0.8) | 0 | 2 (1.0) | 1 |
| R135K | 2 (0.8) | 0 | 2 (1.0) | 1 |
| V136A | 1 (0.4) | 1 (2.0) | 0 | 0.2 |
| R137K/Q | 4 (1.6) | 1 (2.0) | 3 (1.5) | 1 |
| A138D | 1 (0.4) | 1 (2.0) | 0 | 0.2 |
| L139Q/P | 2 (0.8) | 0 | 2 (1.0) | 1 |
| Y140S/N/H/F/C | 9 (3.6) | 2 (4.1) | 7 (3.5) | 1 |
| F141V/L/I | 28 (11.3) | **9 (18.4)** | **19 (9.6)** | **0.08** |
| A143V | 1 (0.4) | 0 | 1 (0.5) | 1 |
| S146F | 2 (0.8) | 0 | 2 (1.0) | 1 |
| S147G | 1 (0.4) | 0 | 1 (0.5) | 1 |
| S148L | 4 (1.6) | 0 | 4 (2.0) | 0.59 |
| G149E/K | 7 (2.8) | 2 (4.1) | 5 (2.3) | 0.63 |
| I150T | 105 (42.5) | 16 (32.7) | 89 (44.9) | 0.12 |
| S152N | 1 (0.4) | 1 (2.0) | 0 | 0.2 |
| P153L | 1 (0.4) | 0 | 1 (0.5) | 1 |
| Q155P/H | 3 (1.2) | 2 (4.1) | 1 (0.5) | 0.1 |
| N156T/I/S | 7 (2.8) | 1 (2.0) | 6 (3.0) | 1 |
| T157S | 2 (0.8) | 0 | 2 (1.0) | 1 |
| V158A | 89 (36.0) | 18 (36.7) | 71 (35.9) | 0.91 |
| A160T/P | 5 (2.0) | 1 (2.0) | 4 (2.0) | 1 |
| I161T/L | 4 (1.6) | 0 | 4 (2.0) | 0.59 |
| T164I/D/S | 15 (6.1) | 3 (6.1) | 12 (6.1) | 1 |
| F165S/L | 14 (5.7) | 3 (6.1) | 11 (5.6) | 1 |
| K167T | 11 (4.5) | 2 (4.1) | 9 (4.5) | 1 |
| T168I | 4 (1.6) | 0 | 4 (2.0) | 0.59 |
| V172A | 3 (1.2) | 1 (2.0) | 2 (1.0) | 0.49 |
| P173Q/L | 4 (1.6) | 1 (2.0) | 3 (1.5) | 1 |
| N174S/K | 2 (0.8) | 1 (2.0) | 1 (0.5) | 0.36 |
| *PreS2* deletion | 40 (16.2) | 11 (22.4) | 29 (14.6) | 0.18 |

percentage by column, [a] Chi-square test.

**Table 4. Distribution of point-mutations on the _S_ gene (n = 247).**

| _S (aa 1–227)_ | Overall population n (%) | HCC n (%) | | p[a] |
|---|---|---|---|---|
| | | Yes (n = 49) | No (n = 198) | |
| **Point-mutations** | | | | |
| E2G | 1 (0.4) | 0 | 1 (0.5) | 1 |
| N3S | 1 (0.4) | 0 | 1 (0.5) | 1 |
| T4I | 1 (0.4) | 0 | 1 (0.5) | 1 |
| A5T/S | 11 (4.5) | 2 (4.1) | 9 (4.5) | 1 |
| F8P | 1 (0.4) | 0 | 1 (0.5) | 1 |
| L9P | 3 (1.2) | 1 (2) | 2 (1) | 0.49 |
| G10R | 2 (0.8) | 1 (2.0) | 1 (0.5) | 0.36 |
| P11H | 1 (0.4) | 0 | 1 (0.5) | 1 |
| L13P/V | 2 (0.8) | 0 | 2 (1.0) | 1 |
| V14A/G/Q | 25 (10.1) | 5 (10.2) | 20 (10.1) | 0.98 |
| L15S | 4 (1.6) | 1 (2) | 3 (1.5) | 1 |
| Q16P | 2 (0.8) | 0 | 2 (1.0) | 1 |
| A17E | 2 (0.8) | 0 | 2 (1.0) | 1 |
| G18R | 2 (0.8) | 1 (2.0) | 1 (0.5) | 0.36 |
| F19S | 1 (0.4) | 1 (2) | 0 | 0.20 |
| F20S | 6 (2.4) | **4 (8.2)** | **2 (1.0)** | **0.015[b]** |
| L21S | 72 (29.1) | 13 (26.5) | 59 (29.8) | 0.65 |
| L22W | 1 (0.4) | 0 | 1 (0.5) | 1 |
| R24K | 4 (1.6) | 0 | 4 (2.0) | 0.59 |
| I25V/A | 4 (1.6) | 0 | 4 (2.0) | 0.59 |
| T27I | 2 (0.8) | 0 | 2 (1.0) | 1 |
| I28T | 1 (0.4) | 0 | 1 (0.5) | 1 |
| Q30R/K | 5 (2.0) | 2 (4.1) | 3 (1.5) | 0.26 |
| S31R | 1 (0.4) | 1 (2) | 0 | 0.20 |
| L32P | 1 (0.4) | 0 | 1 (0.5) | 1 |
| D33G | 2 (0.8) | 2 (4.1) | 0 | **0.039[b]** |
| S34L | 1 (0.4) | 1 (2) | 0 | 0.20 |
| W35STOP | 1 (0.4) | 0 | 1 (0.5) | 1 |
| N40S/K | 17 (6.9) | 2 (4.1) | 15 (7.6) | 0.54 |
| F41S | 2 (0.8) | 0 | 2 (1.0) | 1 |
| L42P/R | 4 (1.6) | 1 (2) | 3 (1.5) | 1 |
| G43K | 1 (0.4) | 0 | 1 (0.5) | 1 |
| G44E/V | 46 (18.6) | 13 (26.5) | 33 (16.7) | 0.11 |
| A45T/G/V | 8 (3.2) | 3 (6.1) | 5 (2.5) | 0.20 |
| P46H/L | 4 (1.6) | 1 (2) | 3 (1.5) | 1 |
| T47A/E/V/K | 23 (9.3) | **9 (18.4)** | **14 (7.1)** | **0.025[b]** |
| C48S | 2 (0.8) | 1 (2) | 1 (0.5) | 0.36 |
| P/L49R/H | 14 (5.7) | 2 (4.1) | 12 (6.1) | 0.74 |
| Q51L | 2 (0.8) | 0 | 2 (1.0) | 1 |
| S53L | 93 (37.7) | 20 (40.8) | 73 (36.9) | 0.61 |
| S59N | 1 (0.4) | 0 | 1 (0.5) | 1 |
| S61L | 10 (4.0) | 2 (4.1) | 8 (4.0) | 1 |
| P62Q/L | 24 (9.7) | 6 (12.2) | 18 (9.1) | 0.51 |
| C64Y | 1 (0.4) | 0 | 1 (0.5) | 1 |
| P67Q | 6 (2.4) | 1 (2) | 5 (2.5) | 1 |

_(Continued)_

**Table 4.** (*Continued*)

| S (aa 1–227) | Overall population n (%) | HCC n (%) | | p[a] |
|---|---|---|---|---|
| | | Yes (n = 49) | No (n = 198) | |
| I68T | 9 (3.6) | 2 (4.1) | 7 (3.5) | 1 |
| R73H | 1 (0.4) | 0 | 1 (0.5) | 1 |
| W74S/L | 6 (2.4) | 0 | 6 (3.0) | 0.60 |
| M75T | 1 (0.4) | 1 (2) | 0 | 0.20 |
| C76Y/T/W | 26 (10.5) | 7 (14.3) | 19 (9.6) | 0.34 |
| L77R | 7 (2.8) | 3 (6.1) | 4 (2.0) | 0.14 |
| R79H | 3 (1.2) | **3 (6.1)** | **0** | **0.007[b]** |
| F80S | 1 (0.4) | 0 | 1 (0.5) | 1 |
| F83S/C | 3 (1.2) | 0 | 3 (1.5) | 1 |
| C85Y/F | 3 (1.2) | 2 (4.1) | 1 (0.5 | 0.1 |
| L88P | 2 (0.8) | **2 (4.1)** | **0** | **0.039[b]** |
| L89P | 1 (0.4) | 0 | 1 (0.5) | 1 |
| I92T | 4 (1.6) | 0 | 4 (2.0) | 0.59 |
| F93S/C | 4 (1.6) | 1 (2) | 3 (1.5) | 1 |
| L95W | 5 (2.0) | 1 (2) | 4 (2.0) | 1 |
| V96G | 1 (0.4) | 1 (2) | 0 | 0.20 |
| L98V | 2 (0.8) | 1 (2.0) | 1 (0.5) | 0.36 |
| Y100C/F | 13 (5.3) | 2 (4.1) | 11 (5.6) | 1 |
| Q101K/H/R | 12 (4.9) | 2 (4.1) | 10 (5.1) | 1 |
| M103T | 2 (0.8) | 0 | 2 (1.0) | 1 |
| L104W | 1 (0.4) | 0 | 1 (0.5) | 1 |
| I110L/Q | 9 (3.6) | 2 (4.1) | 7 (3.5) | 1 |
| R112K | 1 (0.4) | 0 | 1 (0.5) | 1 |
| T113N | 1 (0.4) | 0 | 1 (0.5) | 1 |
| S114T/P/A | 7 (2.8) | 3 (6.1) | 4 (2.0) | 0.14 |
| T115N | 1 (0.4) | 0 | 1 (0.5) | 1 |
| T116N | 1 (0.4) | 1 (2) | 0 | 0.20 |
| T118M | 1 (0.4) | 0 | 1 (0.5) | 1 |
| P120S/T | 21 (8.5) | **8 (16.3)** | **13 (6.6)** | **0.042[b]** |
| R122K | 22 (8.9) | 5 (10.2) | 17 (8.6) | 0.72 |
| T123A/N | 4 (1.6) | 1 (2) | 3 (1.5) | 1 |
| I126T/N/S | 52 (21.1) | 13 (26.5) | 39 (19.7) | 0.29 |
| P127T/A/S | 12 (4.9) | 3 (6.1) | 9 (4.5) | 0.71 |
| A128V | 2 (0.8) | 1 (2) | 1 (0.5) | 0.36 |
| Q129R/N/L/H | 5 (2.0) | 2 (4.1) | 3 (1.5) | 0.26 |
| G130R | 1 (0.4) | 0 | 1 (0.5) | 1 |
| T131N/S | 10 (4.0) | 2 (4.1) | 8 (4.) | 1 |
| S132P | 2 (0.8) | 1 (2) | 1 (0.5) | 0.36 |
| M133T/S/L/I | 19 (7.7) | 5 (10.2) | 14 (7.1) | 0.55 |
| F134Y/V/L/I | 6 (2.4) | 2 (4.1) | 4 (2.0) | 0.34 |
| S136F | 1 (0.4) | 1 (2) | 0 | 0.20 |
| T140I | 7 (2.8) | 1 (2) | 6 (3.0) | 1 |
| T143M | 3 (1.2) | 0 | 3 (1.5) | 1 |
| D144E/A/D | 2 (0.8) | 0 | 2 (1) | 1 |
| G145R/A | 5 (2.0) | **3 (6.1)** | **2 (1)** | **0.055[b]** |
| N147S | 1 (0.4) | 1 (2) | 0 | 0.20 |

(*Continued*)

**Table 4.** (*Continued*)

| S (aa 1–227) | Overall population n (%) | HCC n (%) | | pᵃ |
|---|---|---|---|---|
| | | Yes (n = 49) | No (n = 198) | |
| P151H | 1 (0.4) | 0 | 1 (0.5) | 1 |
| W156L | 2 (0.8) | 1 (2) | 1 (0.5) | 0.36 |
| A159V | 9 (3.6) | 3 (6.1) | 6 (3.0) | 0.39 |
| R160K | 6 (2.4) | 2 (4.1) | 4 (2.0) | 0.34 |
| Y161F/S | 25 (10.1) | 4 (8.2) | 21 (10.6) | 0.79 |
| F162Y | 1 (0.4) | 0 | 1 (0.5) | 1 |
| Y163F | 2 (0.8) | 0 | 2 (1) | 1 |
| E164G | 1 (0.4) | 1 (2.0) | 1 (0.5) | 0.36 |
| A166V/G | 6 (2.4) | 1 (2.0) | 5 (2.5) | 1 |
| S167L | 1 (0.4) | 0 | 1 (0.5) | 1 |
| V168A | 1 (0.4) | 0 | 1 (0.5) | 1 |
| F170S | 1 (0.4) | 1 (2) | 0 | 0.20 |
| W172C | 1 (0.4) | 1 (2) | 0 | 0.20 |
| L173P | 3 (1.2) | 2 (4.1) | 1 (0.5) | 0.10 |
| S174N | 4 (1.6) | **3 (6.1)** | **1 (0.5)** | **0.026**ᵇ |
| L175S | 1 (0.4) | 0 | 1 (0.5) | 1 |
| V177L | 2 (0.8) | 0 | 2 (1) | 1 |
| V180A | 2 (0.8) | 1 (2.0) | 1 (0.5) | 0.36 |
| W182STOP | 1 (0.4) | 0 | 1 (0.5) | 1 |
| A184V/G | 97 (39.3) | 23 (46.9) | 74 (37.4) | 0.22 |
| L186H | 2 (0.8) | 0 | 2 (1) | 1 |
| T189I | 13 (5.3) | 2 (4.1) | 11 (5.6) | 1 |
| V190A | 4 (1.6) | **3 (6.1)** | **1 (0.5)** | **0.026**ᵇ |
| S193L | 4 (1.6) | 0 | 4 (2.0) | 0.59 |
| I195T | 1 (0.4) | 1 (2) | 0 | 0.20 |
| M198I/M | 45 (18.2) | 7 (14.3) | 38 (19.2) | 0.43 |
| W199L/STOP | 2 (0.8) | 1 (2) | 1 (0.5) | 0.36 |
| Y200F/W | 12 (4.1) | 2 (4.1) | 10 (5.1) | 1 |
| P203R | 8 (3.2) | **4 (8.2)** | **4 (2.0)** | **0.052** ᵇ |
| S204R/N | 25 (10.1) | 7 (14.3) | 18 (9.1) | 0.28 |
| L205V | 1 (0.4) | 1 (2) | 0 | 0.20 |
| Y206H/F/C | 6 (2.4) | **3 (6.1)** | 3 (1.5) | **0.094** ᵇ |
| N207S | 1 (0.4) | 1 (2) | 0 | 0.20 |
| I208T/S | 14 (5.7) | 3 (6.1) | 11 (5.6) | 1 |
| L209V/S/G | 5 (2.0) | **3 (6.1)** | **2 (1.0)** | **0.055** ᵇ |
| S210K/N/R/S | 97 (39.3) | 22 (44.9) | 75 (37.9) | 0.42 |
| P211R | 1 (0.4) | 0 | 1 (0.5) | 1 |
| F212Y/L/C | 5 (2.0) | **4 (8.2)** | **1 (0.5)** | **0.006** ᵇ |
| L213I/M | 18 (7.3) | 5 (10.2) | 13 (6.6) | 0.37 |
| L216STOP/Y | 5 (2.0) | 2 (4.1) | 3 (1.5) | 0.26 |
| P217S/L | 2 (0.8) | 1 (2) | 1 (0.5) | 0.36 |
| I218L | 1 (0.4) | 0 | 1 (0.5) | 1 |
| F220Y/L/C | 7 (2.8) | 0 | 7 (3.5) | 0.35 |
| C221Y/R | 11 (4.5) | 1 (2) | 10 (5.1) | 0.70 |
| L222P | 1 (0.4) | 0 | 1 (0.5) | 1 |
| V224A | 29 (11.7) | 7 (14.3) | 22 (11.1) | 0.62 |

(*Continued*)

**Table 4.** (Continued)

| S (aa 1–227) | Overall population n (%) | HCC n (%) | | p^a |
| --- | --- | --- | --- | --- |
| | | Yes (n = 49) | No (n = 198) | |
| Y225F | 1 (0.4) | 1 (2) | 0 | 0.20 |
| I226M/T/S | 5 (2.0) | 0 | 5 (2.5) | 0.59 |
| S Functional sequence (≥1 point-mutations) | | | | |
| MHR | 157 (63.6) | **39 (79.6)** | **118 (59.6)** | **0.009** |
| "a" determinant | 98 (39.7) | 24 (49.0) | 74 (37.4) | 0.14 |

percentage by column,, ^a Chi-square test, ^b Fisher exact test.

specificity (SPE) for predicting HCC respectively, 75%, 81.1%, 6.1%, 99.5%. The predictive values of all other 4 point-mutations related to HCC were concretely described in S3 Table, based on our current available data.

## Discussion

To the best of our knowledge, this investigation was one of the first studies on *PreS/S* gene mutations and their relation with HCC in Vietnamese CHB infected patients. The study population included CHB infected patients with HBV DNA >3 $\log_{10}$-copies/mL (for the higher chance of mutation detection) and had successful *PreS/S* sequencing (for better mutation description and analysis its correlation with HCC). The rate of mutations that were presented in this study therefore might be higher than that of the real HBV infected population in Vietnam. Our study sample composed of 54.7% genotype B, 57.5% HBeAg positive, 23.1% liver fibrosis of >F3, 83% HBV DNA >5 $\log_{10}$-copies/mL and especially 19.8% HCC accompanied. These special characteristics on the population were not only presented the variables that need to be adjusted for their confounding effects but also ensured the aim of detection of mutations and its relationships with HCC.

There were 61.8% amino acid replacements that were detected on the entire *PreS/S* gene. The rates of changes that were higher on the *PreS2* and *S* gene (74.5% of 228 and 70% of 55 amino acid sites, respectively, versus 56.7% of 120 amino acid sites on the PreS1) revealed the high variability of these regions.

On the *PreS1* region that consists of 119 amino acid, 57.1% amino acid replacements were detected with a wide range of mutation rates from 0.4% to 46.9%. However, 79.4% of these replacements (54/68) presented in less than 5% of population. These frequently observed

**Table 5. Point-mutations related to HCC–multivariable analysis (n = 247).**

| Mutation | OR (95%CI) | p |
| --- | --- | --- |
| **W4P/R/Y (*PreS1*)** | 5.48 (1.32–22.83) | 0.019 |
| **F20S (*S*)** | 9.72 (1.55–61.06) | 0.015 |
| **T47A/E/V/K (*S*)** | 2.91 (1.04–8.13) | 0.042 |
| **P120S/T (*S*)** | 4.26 (1.58–11.52) | 0.004 |
| **S174N (*S*)** | 18.21 (1.77–187.65) | 0.015 |
| **P203R (*S*)** | 9.72 (1.55–61.06) | 0.016 |

The characteristics of these 6 point-mutations had been analysed and found that **W4P/R/Y (*PreS1*)** (p = 0.022) and **T47A/E/V/K (*S*)** (p<0.001) had significant higher rates on genotype C; **P120S/T (*S*)** had higher rates on genotype B (p<0.001), HBeAg (-) group (p = 0.019) and low HBV DNA group (<5 $\log_{10}$-copies/mL) (p = 0.013) (S1 Table).

**Table 6. Multivariable analysis for related factors to HCC (n = 247).**

| Variables | OR (95%CI) | P |
|---|---|---|
| **Sex (male)** | 4.51 (1.78–11.4) | 0.001 |
| **Age group (≥40)** | 5.5 (2.06–14.68) | 0.001 |
| **HBeAg negative** | 2.46 (1.1–5.53) | 0.029 |
| **W4P/R/Y (*PreS1*)** | 11.56 (1.99–67.05) | 0.006 |
| **T47A/E/V/K (*S*)** | 3.67 (1.19–11.29) | 0.023 |
| **P120S/T (*S*)** | 3.38 (1.09–10.49) | 0.035 |
| **S174N (*S*)** | 29.73 (2.12–417.07) | 0.012 |
| **P203R (*S*)** | 8.45 (1.43–50.06) | 0.019 |

point-mutations were mostly not related to HCC. Contrarily, 4 point-mutations that belong to the low-rate group were related to HCC. They were W4P/R/Y and S5L/T (p = 0.055) on the NTCP region; A90T/S/G on the HSP70 region and L108V/I on the *S* promoter and B cell epitopes (Table 2).

On the *PreS2* region that consists of 55 amino acids, 74.5% amino acid changes were detected with the mutation rates ranged from 0.4% to 42.5%. 82.9% amino acid changes (34/41) belong to the group of <5% rates. Only F141V/L/I had the higher distribution in the HCC group (18.4% vs 9.6%, p = 0.08) (Table 3). Our finding seemed compatible with a report from Mun *et al.*, who had found that F141L mutation strain increased the risk of HCC in HBV genotype C infected subjects [23]. They had also proved the enhanced cell cycling effects of F141L-expression cell lines through the doubling frequencies of colony-forming versus the wild types.

The *PreS1* deletion (27.5%) and *PreS2* deletion (16.2%) were equally distributed in the HCC and non-HCC groups (Table 4). The same rates of *PreS* deletion (20%) were reported from Matsuo *et al.* (2017) (on 5/21 Vietnamese CHB patients) [18] and from Choi *et al.* in Korean genotype C patients [24]. Literature reviews found mutations of these *PreS* genes effect on retaining of HBV inside the host's cells and on malignant transforming of hepatic cells afterwards [7, 25–27]. More studies had specified on *PreS1* deletion and HCC correlation (Zhang *et al.* (2017) [28], Choi *et al.* (2019) [29]).

On the *S* region, 61.2% amino acid changes were detected with the mutation rates range from 0.4% to 39.3%. We found mutations as Y100C/F, P120S/T, R122K, I126T/N/S, S132P, M133T/S/L/I, G145R (the vaccine escape mutant, 2%), Y200F/W and Y206H/F/C as same as that were reported from other studies (Hudu *et al.* (2015) [30], Kim *et al.* (2018) [20], Hazawa *et al.* (2018) [31], but amino acid change which had been described at amino acid 125 had not been detected in our study.

Moreover from our study, we often found the lower rates of amino acid changes compared to other studies such as from Bui TTT *et al.* (2017) (N38E 71.9%, N38K 71.1%, A60V/E 100% on the *PreS1* region, L126T/S 77% on the *PreS2* and N3S 27.4% on the *S* region) [19], from Kim *et al.* (2018) (K122R 69.3% on the *S* region compared to 8.9% R122K in our study) [20]. Inversely, we detected higher rates of *S* point-mutations as L21S (29.1%), S53L (37.7%), A184V/G (39.3%), S204R/N (10%) and S210K/N/R/S (39.3%), and also on the "a" determinant (39.7% cases with mutation, compared to 7% from Hudu's group [30]. In spite of these lower and higher amino acid change rates, all of these mutations were found not related to HCC in our cross-sectional study. These differences in rates among studies could not only be explained by the distribution of genotype and by the varieties of subgroups in the study populations (such as the co-existence of HBsAg-AntiHBs status, nucleot(s)ide or immunoglobulin treatment, liver cirrhosis and HCC). Moreover, among our study population, antiHBs that had

been tested on 186 cases with clinical symptoms were tested antiHBs and had detected 37 cases (19.9%) with HBsAg-AntiHBs co-exsistence, higher than the rates of 3–5% in other investigation [32, 33]. Therefore, it was presumed that CHB patients with varieties of presentations had been included in our study and contributed to the difference in rates of point-mutations compared to other studies.

We also found a significant higher distribution of cases with mutation on the MHR region (p = 0.009) in the HCC group, especially a higher rate of P120S/T. Outside of the MHR region we also detected higher rates of other 3 *S* mutations T47A/E/V/K, S174N (in the HLA II region) and P203S (in the HLA II region, the C-terminal domain) in the HCC group (Table 4). Hossini *et al*. (2019) had previously found the higher rate of P120T/S in HCC with cirrhosis group [34]. Qiao *et al*. (2017) had also reported that the N-glycosylation mutations on the MHR region accompanied with HBsAg-antiHBs co-existence was related to HCC [35]. Liu *et al*. (2013) further stated that the large N glycosylation of HBsAg antigen modulates HBsAg secretion, causes ER stress, expresses cell cycle and cell proliferation [36].

The mutant strains with amino acid changes at T or B cell epitopes on the *PreS* region can escape the immune surveillance that prolong the HBV infection. Mutants at specific regions of *PreS/S* genes may create premature stop codons, produce abnormal truncated proteins, disbalance the synthesis of surface proteins, result in retaining of HBV inside of the host cells, promote the endoplasmic reticulumn stress pathway, cause DNA oxidative damage and genome instability, upregulate cell cycles and lead to malignant transforming of hepatic cells [7, 25–27]. The *PreS/S* mutant strains enhance cell cycle progression through the down-regulating effects on the p53 and p21 pathways; upregulate the cyclin-dependent kinase 4, cyclin A, hamper HBsAg secretion, increase cellular proliferation [8].

Many other concerns related to the mutation strains and its replacements on virion secretory defect (at amino acid 172 on *S* gene) (Warner *et al*. [37]), on cell proliferation and transformation effects (at amino acids 95, 182, and 216 on S gene) (Huang *et al*. 2014 [38]) or predisposition of the HCC development (at amino acids 69, 95, 182, 216, 210 on S gene) [8, 38]. However, all of these concerned point-mutations were not found related to HCC in our study.

By study on liver tissue of HCC patients (2008), Hatazawa *et al*. had detected 2 *PreS* mutations (W4R and A60V) and more other *PreS* amino acid replacements at codon 5, 30, 35, 5, 54, 77, I84, 98, 102, 118, 123 and 124 [31]. Chen *et al*. (2008) had also reported W4P/R and other changes at codons 7, 81 on the *PreS1*, and at codon 68 on the *S* region related to HCC [39]. Several years later, the significantly higher frequencies of 3 *PreS* mutations at codons 4, 60 and 125 in HCC patients were recorded by Yin *et al*. (2010) [40], Zhang *et al*. [28]. Interestingly in a longitudinal study, Zhang had also observed the increasing of quasi-species complexity and diversity of the HBV strains during the progression to HCC; He had specially stated that the majority of these mutations existed at least 10 years in advance of development of HCC [41]. Zhang *et al*. (2017) had also repeatedly reported significantly higher rates of *PreS* deletion and other *PreS* mutations at codons 4, 27 and 167 in the HCC group [28].

Scientific reviewed on point-mutations that related to HCC, we realized that there were big differences on the patterns and characteristics of the amino acid changes related to HCC between studies. These differences might originate from the structure of study populations, HBV genotypes, the large spectrum of amino acid changes along the *PreS1/PreS2/S* sequences, and the interactions between mutations.

The multivariable analysis was applied twice in our study. Firstly, to adjust interactions between 19 point-mutations that had showed higher rates on the HCC group and recognized 6 mutations which had higher risks of HCC (Table 5). Secondly, to adjust for the confounding effects of personal and viral factors (Table 6). The final findings had recognized 5 mutations

(W4P/R/Y on the *PreS1* region and T47A/E/V/K, P120S/T, S174N, P203R on the *S* region) that significantly related to HCC. The findings that related to the first 3 mutations that were in agreement with other published papers, except the P203R which had not been well reported. Salpini *et al*. (2017) had stated that P203Q and the combination of P203R and S210R hampered the HBsAg secretion and increased cellular proliferation. The correlation of the C-terminus P203Q (17.4% vs 1.0%, p = 0.004), S210R (34.8% vs 3.8%, p<0.001) and of their combination with HCC had been reported in genotype A and D CHB patients [8].

Regarding to the OR values of mutations on the final multivariable analysis, 2 *S* mutations including 23 cases of **T47A/E/V/K** (*S*) and 21 cases of **P120S/T** (*S*) had revealed three folds increase in HCC risk associated with reasonable confidential intervals. On the contrary, three remaining mutations had only been detected on small numbers of cases with especially high ORs and wide 95% confidential intervals including 3 cases of **W4P/R/Y** (*preS1)*, OR 11.56 (1.99–67.05); 4 cases of **S174N** (*S*), OR 29.73 (2.12–417.07); and 8 cases of **P203R** (*S*), OR 8.45 (1.43–50.06) (Table 6). A small sample size of this study resulted in a wider confidence interval with a larger margin of error for these sporadic mutations. It was suggested that a tighter confidence interval with values closer to the actual OR would be obtained if the sample size was increased. We had calculated the predictive values of these 5 point-mutations and had all found the high SPE and NPV values, but all revealed modest SEN due to small number of cases. S174N (*S*) for instance had been observed in our study with the highest OR and relative high PPV, NPV and specificity (75%, 81.1% and 99.5%, respectively) but its sensitivity was only 6.1% (S3 Table). If possible, the deep sequencing technique with its higher sensitivity could either potentially increase detection rates or improve the SEN values of these low frequency point-mutations. However, we were unable to perform it this time due to a large cost associated with the technique. Further studies are recommended in continuing upon findings of this study in which the direct sequencing would be the best and compulsory technique for better recognizing point mutations at quacispecies levels.

Contrarily, frequent amino acid replacements in our study were detected at the widely known structural and functional sequences such as N51Y/T/S/Q (30.4%), V68T/S/I (44.9%) on the *S* promoter; T/N87S/T/P (46.2%) on the HSP 70 (heat shock protein) and T125S/N/P (30.8%) on the NBS region. These structures are concerned by their role on the structure and morphology of HBV, the dual topology of L proteins (HSP70), the CAD—Cytosolic anchorage determinant), the virion morphogenesis (NBS—*The nucleocapside binding site*) and the S RNA transcription (*The S promoter and the CCAAT/CBF*) [42]. At the cellular level, the mutations at these functional regions has been known to contribute to the production and secretion of surface proteins, on the intracellular retention of envelope proteins and on the endoplasmic reticulum (ER) stress [43]. However, these above point-mutations had equally distributed on the HCC and non-HCC group in our study. More longitudinal cohorts need to be continued apart from this population because the diseases and HCC outcomes need at least one or more decades to appear.

There were some HCC related factors that were not included in the multivariable analysis such as Basal core promoter mutations, history of vaccination, HBsAg-antiHBs co-existence status, HBV genotype, cirrhotic status. Also, the combination of mutations and their interactive effects had not yet been analyzed. Other limitations of our study were also rooted from the study population that was not large enough for the low-rate mutations. A wide spectrum of significant mutations on the 3 regions (*PreS1*, *PreS2* and *S*) and the interactive effects between mutations that need to be concretely clarified.

Further larger investigation and observation longitudinal studies were in need to be done to describe and analyse the relation between *PreS/S* mutation and HCC.

## Conclusions

61% amino acid changes with a broad range of mutation rates were detected on the *PreS1/ PreS2/S* regions of chronic HBV infected patients. The W4P/R/Y (on *preS1* region) and T47A/ E/V/K, P120S/T, S174N and P203R (on *S* region) were found related to HCC. Further investigation included cohort studies are recommended to continue to further investigate the relation of mutations on the HBV genome and HCC outcome.

## Supporting information

**S1 Table. Distributions of 6 mutations related to HCC in groups of personal and HBV characteristics.**
(DOCX)

**S2 Table. Distribution of personal characteristics and HBV viral markers in HCC and non HCC group (n = 247).**
(DOCX)

**S3 Table. Predictive values of point-mutations related to HCC (n = 247).**
(DOCX)

**S1 File.**
(XLSX)

## Acknowledgments

We would like to acknowledge all patients who participated in this study. Special thanks all the medical health staff members of Hepatology Clincic of UMC and Center for Molecular Biomedicine of UMP at Ho Chi Minh city who attended in patient recruitment, blood sampling, storage, sequencing and interpreting the results.

## Author Contributions

**Conceptualization:** Nguyen Thi Cam Huong, Nguyen Quang Trung, Bac An Luong, Hoang Anh Vu, Hoang Huu Bui, Hoa Pham Thi Le.

**Data curation:** Nguyen Thi Cam Huong, Bac An Luong, Hoang Anh Vu, Hoa Pham Thi Le.

**Formal analysis:** Nguyen Thi Cam Huong, Nguyen Quang Trung, Bac An Luong, Duong Bich Tram, Hoang Anh Vu, Hoa Pham Thi Le.

**Funding acquisition:** Nguyen Thi Cam Huong.

**Investigation:** Nguyen Thi Cam Huong, Nguyen Quang Trung, Hoang Huu Bui, Hoa Pham Thi Le.

**Methodology:** Nguyen Thi Cam Huong, Nguyen Quang Trung, Bac An Luong, Hoang Anh Vu, Hoa Pham Thi Le.

**Project administration:** Nguyen Thi Cam Huong, Hoa Pham Thi Le.

**Software:** Bac An Luong, Duong Bich Tram.

**Supervision:** Nguyen Thi Cam Huong, Hoa Pham Thi Le.

**Validation:** Nguyen Thi Cam Huong, Hoang Anh Vu, Hoa Pham Thi Le.

**Visualization:** Nguyen Thi Cam Huong, Hoang Anh Vu, Hoa Pham Thi Le.

**Writing – original draft:** Nguyen Thi Cam Huong, Nguyen Quang Trung, Bac An Luong, Hoang Anh Vu, Hoang Huu Bui, Hoa Pham Thi Le.

**Writing – review & editing:** Nguyen Thi Cam Huong, Nguyen Quang Trung, Bac An Luong, Hoang Anh Vu, Hoang Huu Bui, Hoa Pham Thi Le.

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
