## [Decision Letter · Decision Letter 0]

18 Nov 2021

PONE-D-21-33925Mutations in the HBV PreS/S gene related to hepatocellular carcinoma in Vietnamese chronic HBV-infected patientsPLOS ONE

Dear Dr. Thi Cam Huong,

Thank you for submitting your manuscript to PLOS ONE. After careful consideration, we feel that it has merit but does not fully meet PLOS ONE’s publication criteria as it currently stands. Therefore, we invite you to submit a revised version of the manuscript that addresses the points raised during the review process.

Four independent scientists in the field together with editor reviewed this paper, all judged the importance of the findings obtained, however, some major modifications are still required especially, since S174N (S) point mutation showed the highest OR with significant P value, what are the sensitivity, specificity, PPV and NPV for this point mutation. It is also recommended to deep sequencing on some samples to see the rate of quasispecies of some of these significant mutations. 

We look forward to receiving your revised manuscript.

Kind regards,

Hussein Hassan Aly, PhD, MD

Academic Editor

PLOS ONE

Journal Requirements:

Additional Editor Comments:

4 independent reviewers together with editor reviewed this paper, all judged the importance of the findings obtained, however, the some major modifications are required especially, since S174N (S) point mutation showed the highest OR with significant P value, what are the sensitivity, specificity, PPV and NPV for this point mutation. It is also recommended to deep sequencing on some samples to see the rate of quasispecies of some of these significant mutations

Reviewers' comments:

Reviewer's Responses to Questions

**Comments to the Author**

1. Is the manuscript technically sound, and do the data support the conclusions?

Reviewer #1: Yes

Reviewer #2: Partly

Reviewer #3: Yes

Reviewer #4: Yes

2. Has the statistical analysis been performed appropriately and rigorously? 

Reviewer #1: Yes

Reviewer #2: Yes

Reviewer #3: Yes

Reviewer #4: Yes

3. Have the authors made all data underlying the findings in their manuscript fully available?

Reviewer #1: Yes

Reviewer #2: Yes

Reviewer #3: Yes

Reviewer #4: Yes

4. Is the manuscript presented in an intelligible fashion and written in standard English?

Reviewer #1: Yes

Reviewer #2: No

Reviewer #3: No

Reviewer #4: Yes

5. Review Comments to the Author

Reviewer #1: My concerns regarding this manuscript are:

1-The history of HB vaccination and HBsAb status of the studied patients are not presented here, indeed the occurrence of Pre-S/S gene mutations may be affected by both factors.

2- The duration of HBV infection of included patients and its relation to pre-S/S mutations as well as the HCC occurrence should be shown and analyzed.

3- The method of liver fibrosis assessment is not mentioned in methodology, this should be clarified

4- It would be of clinical interest; to present the relation between liver inflammation status on one hand (e.g., ALT, AST or grade of liver inflammation on liver biopsy if available) and the pre-s/s mutations as well as HCC occurrence on the other hand

Reviewer #2: In the manuscript entitled "Mutations in the HBV PreS/S gene related to hepatocellular carcinoma in Vietnamese chronic HBV-infected patients", the authors performed a cross-sectional study including Vietnamese CHB-infected patients with high HBV-DNA >and performed PreS/S genes sequencing. The article is well presented despite the need for vigorous language revisions. My main concern is the value of generalising the presenting data which is limited to the Vietnamese population.

Reviewer #3: PONE-D-21-33925

The strong point of this manuscript

-Aim of the study is good and important

-good results

Weakness of the manuscript

1-Englsih editing is required

The follaoing is my comments

Abstract

Please add the following paragraph in new Materials section

Ine 35-35 page 8

247 CHB-infected patients were recruited, characterized by 68.8% males, 44.5%

HBV genotype C, 54.4% HBeAg positive, 23.1% fibrosis score ≥F3 and 19.8% HCC

Please add major hydrophilic region line 39 page 8 (with point-mutation on major hydrophilic region (MHR)

Please delete "We detected a wide spectrum of point-mutations on PreS/S regions in

Vietnamese CHB patients line" 48 page 8

Introduction

Please correct the reference Wang et al (2020)

Please correct the reference Dunford et al group (2012)

Materiasl and methods

Is ok

Statistics is included

Ethics Considerations is included

Results

The presentation of the results is more complicated and it is better if authors represented it with more simple figures

Discussion

Please delete the subtitles in the discussion section

There are several of references are not accrding to the style of the journal such as

(2017) [27], Choi et al (2019), [28]).

(Hudu el al 2015 [29], Kim et al 2018 [20], Hazawa 2018 [30],

Bui TTT et al (2017)

Kim et al (2018)

Qiao et. al (2017)

Liu et al (2013)

(Warner et al [34]),

(Huang 2014

(2008), Hatazawa et al

Chen et al (2008)

Yin et al (2010)

Please delete Recently, and correct the reference Zhang et al (2017)

References

Contains several recent references 2020, 2021

Reviewer #4: Huong, et al. investigated the correlation between preS/S of HBV genome mutations and HCC in HBV-Vietnamese patients. The study is interesting, however, there are some issues need to be addressed:

1- Is there any significant correlation between the mutations identified in the multivariable analysis in table 6 and sex, age and HBeAg negative.

2- Since S174N (S) point mutation showed the highest OR with significant P value, what are the sensitivity, specificity, PPV and NPV for this point mutation.

3- It is recommended to deep sequencing on some samples to see the rate of quasispecies of some of these significant mutations.

4- Tables 2,3 and 4 are very long, might move as supplementary tables.

5- In the abstract (at the end of third line of the results), it looks there are some information is missing here "and on PreS1 gene"

6- Any abbreviations should be spelled out in the first place, such as MHR in the abstract.

6. PLOS authors have the option to publish the peer review history of their article (what does this mean?). If published, this will include your full peer review and any attached files.

Reviewer #1: No

Reviewer #2: No

Reviewer #3: No

Reviewer #4: No

---

## [Author Response · Author response to Decision Letter 0]

16 Jan 2022

Response to reviewers

We would like to express our gratitude to the editors and reviewers of this manuscript for thorough review and providing us with highly constructive comments to improve it. We have revised the manuscript according to the suggestions and made appropriate changes where needed.

Editor's comments:

• Comments: We note that you have included the phrase “data not shown” in your manuscript. Unfortunately, this does not meet our data sharing requirements. PLOS does not permit references to inaccessible data. We require that authors provide all relevant data within the paper, Supporting Information files, or in an acceptable, public repository. Please add a citation to support this phrase or upload the data that corresponds with these findings to a stable repository (such as Figshare or Dryad) and provide and URLs, DOIs, or accession numbers that may be used to access these data. Or, if the data are not a core part of the research being presented in your study, we ask that you remove the phrase that refers to these data.

Reply: We removed the phrase “data not shown” in line 230, page 22, and replaced with data was shown completely in Table S2 "Distribution of personal characteristics and HBV viral markers in HCC and non HCC group". All relevant data are within the manuscript and its supporting information as File S2_Pres_S mutation related to HCC.xlxs.

• Comments: 4 independent reviewers together with editor reviewed this paper, all judged the importance of the findings obtained, however, the some major modifications are required especially, since S174N (S) point mutation showed the highest OR with significant P value, what are the sensitivity, specificity, PPV and NPV for this point mutation. It is also recommended to deep sequencing on some samples to see the rate of quasispecies of some of these significant mutations.

Reply: Regarding to the OR values of mutations on the final multivariable analysis, 2 S mutations (T47A/E/V/K(S) and P120S/T(S)) had revealed three fold increase in HCC risk and good confidential intervals. On the contrary, three remaining mutations had only been detected on small numbers of cases (W4P/R/Y(preS1) 3 cases, OR 11.56 (1.99-67.05); S174N(S) 4 cases, 29.73 (2.12-417.07); and P203R (S) 8 cases, OR 8.45 (1.43-50.06)) with especially high ORs (table 6). The wide 95% confidential intervals of OR with very high upper values of these intervals revealed that the sample size was not large enough for investigating these sporadic mutations. Ideally, the deep sequencing technique could potentially increase the detection rates due to its higher sensitivity. However, the cost effectiveness should be considered for the aims of studying these irregular mutations or investigating a larger population. We mentioned these explanations in line 350-360. Due to limited condition, we were unable to perform deep sequencing in this study to determine sensitivity, specificity, PPV and NPV for this point mutation S174N (S).

Reviewer #1 's opinions:

• Comments: The history of HB vaccination and HBsAb status of the studied patients are not presented here, indeed the occurrence of Pre-S/S gene mutations may be affected by both factors.

Reply: Thank you for your great feedback.

During sample collection, the history of HB vaccination was not recorded in 212 CHB patients whose serum samples had been stored during 2014-2016. It was only recorded in 35 CHB patients that were recruited in 2020-2021. Thus, there was no sufficient data available in all patients that can be utilized to analyze the relation of the HB vaccination to Pre-S/S gene mutations.

In line 294-299, we has clarified: "Moreover, among our study population, antiHBs that had been tested on 186 cases with clinical symptoms were tested antiHBs and had detected 37 cases (19.9%) with HBsAg-AntiHBs coexsistence, higher than the rates of 3-5% in other investigation. Therefore, it was presumed that CHB patients with varieties of presentations had been included in our study and contributed to the difference in rates of point-mutations compared to other studies".

• Comments: The duration of HBV infection of included patients and its relation to pre-S/S mutations as well as the HCC occurrence should be shown and analyzed.

Reply: Data from hospital electronic database cannot be used to directly respond to this question. Alternatively, the time from when patients had been diagnosed with CHB to the time of study participation was calculated using data from “year of CHB diagnosed” and “year of inclusion”.

• Comments: The method of liver fibrosis assessment is not mentioned in methodology, this should be clarified.

Reply: In line 117-123, it is added that "Cirrhosis was defined as having signs of portal hypertension (splenomegaly, ascites, vascular collaterals on abdominal wall, esophageal varices or portal hypertensive gastropathy on gastroscopy) and signs of insufficiency of liver function (palmar erythema, vascular spiders, low concentration of albumin (<35 g/dL), elevation of the international normalized ratio (INR >1.1), thrombocytopenia (<160,000/mm3)) as well as irregularity of hepatic surface on ultrasonography or F3 on Metavir score using Acoustic radiation force impulse (ARFI) measurement [22]". We based on the reference of "Udell JA, et al. Does this patient with liver disease have cirrhosis? JAMA. 2012;307(8):832-42".

• Comments: It would be of clinical interest; to present the relation between liver inflammation status on one hand (e.g., ALT, AST or grade of liver inflammation on liver biopsy if available) and the pre-s/s mutations as well as HCC occurrence on the other hand

Reply: In our data, we only used ALT cross-sectional data (one time at inclusion) and we didn't have information on liver biopsy. Relation between ALT (non repeated measurementy) and mutation and HCC were not in the scope of this study. 

Reviewer #2 's opinions:

• Comments: My main concern is the value of generalising the presenting data which is limited to the Vietnamese population.

Reply: Thank you for your comment. With advantage of having decent number of patients in different stages of CHB enrolled in the study, our data can moderately represent the Vietnamese population. However, as the study was only conducted on patients with detectable HBV DNA, the rates of mutations are predicted to be possibly higher than general population. Therefore, in the scope of this study, we do not aim to generalize the frequencies of mutation. Instead, it is intended to determine the correlation between mutation of HBV on PreS/S genes and HCC in vietnamese population. Further studies are suggested to generalize data to chronic HBV infected HCC community.

Reviewer #3 opinions: Many thanks for such addition

• Comments: English editing is required

Reply: We edited academic English writing whole our manuscript.

• Comments: Please add the following paragraph in new Materials section

Iine 35-35 page 8 247 CHB-infected patients were recruited, characterized by 68.8% males, 44.5% HBV genotype C, 54.4% HBeAg positive, 23.1% fibrosis score ≥F3 and 19.8% HCC

Reply: It was summarized in the Result section.

• Comments: Please add major hydrophilic region line 39 page 8 (with point-mutation on major hydrophilic region (MHR) in abstract.

Reply: we edited “point-mutation on major hydrophilic region (MHR)” in abstract line 42.

• Comments: Please delete "We detected a wide spectrum of point-mutations on PreS/S regions in

Vietnamese CHB patients” line 48 page 8

Reply: We removed that sentence and replaced with " We detected 61% amino acid changes on PreS/S regions in Vietnamese CHB patients" line 50-51, page 2. 

• Comments: There are several of references are not according to the style of the journal

Reply: we have completed a thorough review of all the references used in our study and made changes accordingly 

• Comments: The presentation of the results is more complicated and it is better if authors represented it with more simple figures.

Reply: The tables are intended to describe the name of point-mutations along with their associated data for traceability purpose.

• Comments: In discussion: Please delete the subtitles in the discussion section, delete "Recently"

Reply: all subtitles and "Recently" are deleted as per comment.

Reviewer #4 's opinions:

• Comments: Is there any significant correlation between the mutations identified in the multivariable analysis in table 6 and sex, age and HBeAg negative.

Reply: We have completed the data analysis and revised the result line 221-224 and Table S1: The characteristics of these 6 point mutations had been analysed and found that W4P/R/Y (PreS1) (p=0.022) and T47A/E/V/K (S) (p<0.001) had significant higher rates on genotype C; P120S/T (S) had higher rates on genotype B (p<0.001), HBeAg (-) group (p=0.019) and low HBV DNA group (<5 log copies/mL) (p=0.013) (Table S1).

• Comments: Since S174N (S) point mutation showed the highest OR with significant P value, what are the sensitivity, specificity, PPV and NPV for this point mutation. It is recommended to deep sequencing on some samples to see the rate of quasispecies of some of these significant mutations (as above).

• Comments: Tables 2,3 and 4 are very long, might move as supplementary tables.

Reply: Tables are kept as it to support the clarification of data collected and considered critical to overall results.

• Comments: In the abstract (at the end of third line of the results), it looks there are some information is missing here "and on PreS1 gene"

Reply: We have added in line 38-40: The most common point mutations included N/H51Y/T/S/Q/P (30.4%), V68T (44.9%), T/N87S/T/P (46.2%) on PreS1 gene.

• Comments: Any abbreviations should be spelled out in the first place, such as MHR in the abstract.

Reply: "major hydrophilic region (MHR)" is added in line 42.

Sincerely, 

Nguyen Thi Cam Huong

MD, PhD, Lecturer

Departments of Infectious diseases

University of Medicine and Pharmacy at Ho Chi Minh city

---

## [Decision Letter · Decision Letter 1]

28 Feb 2022

PONE-D-21-33925R1Mutations in the HBV PreS/S gene related to hepatocellular carcinoma in Vietnamese chronic HBV-infected patientsPLOS ONE

Dear Dr. Thi Cam Huong,

Thank you for submitting your manuscript to PLOS ONE. After careful consideration, we feel that it has merit but does not fully meet PLOS ONE’s publication criteria as it currently stands. Therefore, we invite you to submit a revised version of the manuscript that addresses the points raised during the review process.

The manuscript has been significantly improved, however, the authors failed to comply with the reviewer comment regarding the calculation of PPV, NPV, sensitivity and specificity using deep sequencing or their available current data. This would give a clinical significance for their findings.

We look forward to receiving your revised manuscript.

Kind regards,

Hussein H. Aly, PhD, MD

Academic Editor

PLOS ONE

Reviewers' comments:

Reviewer's Responses to Questions

**Comments to the Author**

1. If the authors have adequately addressed your comments raised in a previous round of review and you feel that this manuscript is now acceptable for publication, you may indicate that here to bypass the “Comments to the Author” section, enter your conflict of interest statement in the “Confidential to Editor” section, and submit your "Accept" recommendation.

Reviewer #1: All comments have been addressed

Reviewer #3: All comments have been addressed

Reviewer #4: (No Response)

2. Is the manuscript technically sound, and do the data support the conclusions?

Reviewer #1: Yes

Reviewer #3: Yes

Reviewer #4: Partly

3. Has the statistical analysis been performed appropriately and rigorously? 

Reviewer #1: Yes

Reviewer #3: Yes

Reviewer #4: No

4. Have the authors made all data underlying the findings in their manuscript fully available?

Reviewer #1: Yes

Reviewer #3: Yes

Reviewer #4: Yes

5. Is the manuscript presented in an intelligible fashion and written in standard English?

Reviewer #1: Yes

Reviewer #3: Yes

Reviewer #4: No

6. Review Comments to the Author

Reviewer #1: Dear authors thanks for your response , all comments have been addressed , I have no further comments regarding this work

Reviewer #3: The authors did all my revised comments, particularly in English editing, reference style, and deleting words recently from some sentences, and I accepted them in this form.

Reviewer #4: (No Response)

7. PLOS authors have the option to publish the peer review history of their article (what does this mean?). If published, this will include your full peer review and any attached files.

Reviewer #1: No

Reviewer #3: **Yes: **Ashraf A.Tabll

Reviewer #4: No

---

## [Author Response · Author response to Decision Letter 1]

6 Mar 2022

Response to reviewers

We would like to express our gratitude to the editors and reviewers for recognizing our effort to improve our manuscript. We have revised the manuscript according to the suggestions and made appropriate changes where needed.

Comments: "The manuscript has been significantly improved, however, the authors failed to comply with the reviewer comment regarding the calculation of PPV, NPV, sensitivity and specificity using deep sequencing or their available current data. This would give a clinical significance for their findings”.

Reply: It has been described in line 239-243 of the result section: "S174N (S) had the highest OR (29.73) with positive predictive value (PPV), negative predictive value (NPV), sentitivity (SEN), specificity (SPE) for predicting HCC respectively, 75%, 81.1%, 6.1%, 99.5%. The predictive values of all other 4 point-mutations related to HCC were concretely described in Table S3, based on our current available data."

We explained more in line 354-372 of discussion section: "Regarding to the OR values of mutations on the final multivariable analysis, 2 S mutations including 23 cases of T47A/E/V/K (S) and 21 cases of P120S/T (S) had revealed three folds increase in HCC risk associated with reasonable confidential intervals. On the contrary, three remaining mutations had only been detected on small numbers of cases with especially high ORs and wide 95% confidential intervals including 3 cases of W4P/R/Y (preS1), OR 11.56 (1.99-67.05); 4 cases of S174N (S), OR 29.73 (2.12-417.07); and 8 cases of P203R (S), OR 8.45 (1.43-50.06)) (table 6). A small sample size of this study resulted in a wider confidence interval with a larger margin of error for these sporadic mutations. It was suggested that a tighter confidence interval with values closer to the actual OR would be obtained if the sample size was increased. We had calculated the predictive values of these 5 point-mutations and had all found the high SPE and NPV values, but all revealed modest SEN due to small number of cases. S174N (S) for instance had been observed in our study with the highest OR and relative high PPV, NPV and specificity (75%, 81.1% and 99.5 %, respectively) but its sensitivity was only 6.1% (Table S3). If possible, the deep sequencing technique with its higher sensitivity could either potentially increase detection rates or improve the SEN values of these low-frenquency point-mutations. However, we were unable to perform it this time due to a large cost associated with the technique. Further studies are recommended in continuing upon findings of this study in which the direct sequencing would be the best and compulsory technique for better recognizing point mutations at quacispecies levels."

Sincerely, 

Nguyen Thi Cam Huong

MD, PhD, Lecturer

Departments of Infectious diseases

University of Medicine and Pharmacy at Ho Chi Minh city, Vietnam

---

## [Editor Report · Decision Letter 2]

15 Mar 2022

Mutations in the HBV PreS/S gene related to hepatocellular carcinoma in Vietnamese chronic HBV-infected patients

PONE-D-21-33925R2

Dear Dr. Thi Cam Huong,

We’re pleased to inform you that your manuscript has been judged scientifically suitable for publication and will be formally accepted for publication once it meets all outstanding technical requirements.

Kind regards,

Hussein H. Aly, PhD, MD

Academic Editor

PLOS ONE

Additional Editor Comments (optional):

The Authors significantly improved the manuscript and efficiently answered reviewers' comments
---

## [Editor Report · Acceptance letter]

22 Mar 2022

PONE-D-21-33925R2 

Mutations in the HBV PreS/S gene related to hepatocellular carcinoma in Vietnamese chronic HBV-infected patients 

Dear Dr. Thi Cam Huong:

I'm pleased to inform you that your manuscript has been deemed suitable for publication in PLOS ONE. Congratulations! Your manuscript is now with our production department. 

Kind regards, 

on behalf of

Dr. Hussein H. Aly 

Academic Editor

PLOS ONE